# Fabrication and Characterization of Dielectric ZnCr_2_O_4_ Nanopowders and Thin Films for Parallel-Plate Capacitor Applications

**DOI:** 10.3390/mi14091759

**Published:** 2023-09-12

**Authors:** Vasyl Mykhailovych, Gabriel Caruntu, Adrian Graur, Mariia Mykhailovych, Petro Fochuk, Igor Fodchuk, Gelu-Marius Rotaru, Aurelian Rotaru

**Affiliations:** 1Department of Electrical Engineering and Computer Science & Research Center MANSiD, Stefan cel Mare University of Suceava, 13, University St., No. 13, 720229 Suceava, Romania; vasyl.mykhailovych@usm.ro (V.M.); adriang@usm.ro (A.G.); m.mykhailovych@gmail.com (M.M.); 2Department of General Chemistry and Material Science, Yuriy Fedkovych Chernivtsi National University, 2, Kotsjubynskyi St., 58012 Chernivtsi, Ukraine; fochukp@gmail.com; 3Physical, Technical and Computer Science Institute, Yuriy Fedkovych Chernivtsi National University, 2, Kotsjubynskyi St., 58012 Chernivtsi, Ukraine; 4Department of Chemistry and Biochemistry, Central Michigan University, 1200 S. Franklin St., Mount Pleasant, MI 48859, USA; 5Science of Advanced Materials Program, Central Michigan University, 1200 S. Franklin St., Mount Pleasant, MI 48859, USA; 6Faculty of Mechanical Engineering Mechatronics and Management & Research Center MANSiD, Stefan cel Mare University, 720229 Suceava, Romania; gelu.rotaru@usm.ro

**Keywords:** high-k material, ZnCr_2_O_4_ nanoparticles, shape-controlled synthesis, dielectric properties, thin films, planar capacitor

## Abstract

We report here the successful shape-controlled synthesis of dielectric spinel-type ZnCr_2_O_4_ nanoparticles by using a simple sol-gel auto-combustion method followed by successive heat treatment steps of the resulting powders at temperatures from 500 to 900 °C and from 5 to 11 h, in air. A systematic study of the dependence of the morphology of the nanoparticles on the annealing time and temperature was performed by using field effect scanning electron microscopy (FE-SEM), powder X-ray diffraction (PXRD) and structure refinement by the Rietveld method, dynamic lattice analysis and broadband dielectric spectrometry, respectively. It was observed for the first time that when the aerobic post-synthesis heat treatment temperature increases progressively from 500 to 900 °C, the ZnCr_2_O_4_ nanoparticles: (i) increase in size from 10 to 350 nm and (ii) develop well-defined facets, changing their shape from shapeless to truncated octahedrons and eventually pseudo-octahedra. The samples were found to exhibit high dielectric constant values and low dielectric losses with the best dielectric performance characteristics displayed by the 350 nm pseudo-octahedral nanoparticles whose permittivity reaches a value of ε = 1500 and a dielectric loss tan δ = 5 × 10^−4^ at a frequency of 1 Hz. Nanoparticulate ZnCr_2_O_4_-based thin films with a thickness varying from 0.5 to 2 μm were fabricated by the drop-casting method and subsequently incorporated into planar capacitors whose dielectric performance was characterized. This study undoubtedly shows that the dielectric properties of nanostructured zinc chromite powders can be engineered by the rational control of their morphology upon the variation of the post-synthesis heat treatment process.

## 1. Introduction

The continuous advances in microelectronics and computing require the development and improvement of active elements of electronic circuitry. Spinel type materials have been the workhorse in electronics due to their high stability, unique tunable magnetic and electric properties, along with easy processability [1]. Unlike their ferrite counterparts, transition metal chromites adopt, in bulk, the normal spinel structure, due to the large crystal field stabilization energy of the Cr^3+^ ions crystallizing in the cubic system (space group *F*$\bar{d3}$*m*, No. 227). [2] Among chromites, zinc chromite (ZnCr_2_O_4_) has emerged as an interesting catalytic material in the oxidation of CO, the catalytic combustion of hydrocarbons and others [3,4,5], synthesis of methanol [6], photocatalysis [7,8], sensing [9,10,11,12], the design of near-infrared spectral emitters [13,14], and broadband photo-detectors [15]. Moreover, ZnCr_2_O_4_ is a wide bandgap semiconductor whose energy gap varies between 3.0 and 3.5 eV, depending on the size of the constituting particles and the existence of structural defects [11,16,17], which makes it a potential dielectric material for applications in micro- and nanoelectronics. Interestingly, bulk zinc chromite is a typical spin-frustrated material that undergoes an antiferromagnetic transition at the Néel temperature T_N_ = 12 K. Kagomyia and colleagues demonstrated that the relaxation of this magnetic frustration is lattice-mediated, leading to the onset of an anomaly in the variation of the dielectric permittivity at the Néel temperature along with a dispersion of the dielectric permittivity at temperatures below 70 K [18]. In recent years, extensive efforts have been devoted to the investigation of the dielectric properties of both bulk and nanostructured spinel ZnCr_2_O_4_ [18,19,20,21]. Javed et al. [19] investigated the dielectric properties of ZnCr_2_O_4_ nanoparticles with an average size of 144 nm, obtained by the sol-gel auto-combustion method, showing that the dielectric constant varied from ε = 44 to ε = 20 within a frequency range between 70 Hz and 1 MHz. Similarly, Shafqat et al. [22] investigated the structural, morphological and dielectric properties of nanoscale spinel transition metal chromites XCr_2_O_4_ (X = Zn, Mn, Cu and Fe), finding that the permittivity of pelletized powders is the highest for ZnCr_2_O_4_ nanoparticles, ranging from ε = 100,000 to ε = 100 for frequencies varying between 20 Hz and 20 MHz, respectively. In an impedance analysis study of nanostructured zinc chromite by Naz et al. [20], the dielectric permittivity of 50 nm nanoparticles synthesized through a hydrothermal route was found to decrease from ε = 40,000 to ε = 7 in the frequency range between 1 Hz and 10 MHz. The comprehensive assessment of the dielectric permittivity of nanoparticles generally relies on various factors, including the crystal structure, morphology and porosity. These three main parameters significantly impact the results of the dielectric spectroscopy measurements; hence a step-by-step and thorough investigation of each parameter is necessary for the reliable evaluation of the dielectric properties of these materials. The determination of the structure and the purity of spinel ZnCr_2_O_4_ nanopowders can be achieved through detailed X-ray analysis. Additionally, valuable structural information can be obtained using Raman and Energy Dispersive X-ray analyses. Once the spinel structure is determined, the next parameter to consider is morphology. The morphology of particles plays a significant role in dielectric investigation as it directly impacts the properties of the material and the porosity of the final product. 

A noteworthy study is that of Binks et al., who simulated the crystal morphology of ZnCr_2_O_4_ ceramic-predicted four morphologies of cubic spinel lattice ZnCr_2_O_4_ crystallites, namely: octahedral, slightly trunked octahedral in vertices, cubic and heavily capped octahedron [23]. Furthermore, based on surface energy considerations, they also calculated that the main crystallite growth of ZnCr_2_O_4_ occurs into a regular octahedral geometry with the (111) crystallographic plane dominating. However, it is important to note that the morphology of spinel-type oxides can be influenced by several factors. For example, Xiao and co-authors demonstrated a high level of control over the morphology of spinel Co_3_O_4_ nanoparticles. By adjusting the ratio of cobalt nitrate hexahydrate and co-precipitant sodium hydroxide during hydrothermal synthesis, they were able to obtain cubic, truncated octahedral and octahedral shapes [24]. This clearly indicates that the surface or the attachment energies can be influenced by the precursor concentration and reaction condition, ultimately dictating the morphology of the nanoparticles. In a different study, Parhi et al. synthesized ZnCr_2_O_4_ nanoparticles with similar octahedral morphology as predicted by Binks and colleagues [23]. They synthesized the nanoparticles using the microwave metathetic approach, which predominantly led to the formation of an octahedral morphology. Similarly, Mancic et al. investigated ZnCr_2_O_4_ nanoparticles synthesized through the aerosol reaction of precursors, considering their stoichiometry and morphology [25]. Interestingly, the stoichiometry of the particles was found to be influenced by the aerosol residence time (3, 6 and 9 s at 700 °C), while an additional annealing at 1000 °C led to the transformation from pseudospherical shape to a mainly octahedral shape of the particles. These studies highlight the existence of several approaches to change or control the morphology of ZnCr_2_O_4_ nanoparticles, such as using different synthesis methods or varying the reaction conditions. 

However, an investigation of ZnCr_2_O_4_ nanoparticle morphology changes until the complete formation of the spinel structure and its dependence on the thermal treatment history of the samples has not been previously conducted. Therefore, we investigated the structure, morphology progress and dielectric properties of ZnCr_2_O_4_ nanoparticles as a function of annealing temperature and time. The main goal was to find the optimal conditions that allow for a reliable evaluation of the dielectric constant for nanoparticles obtained at different temperatures. Additionally, we fabricated a series of thin films and a high-k capacitor using a thin film of ZnCr_2_O_4_ nanoparticles that exhibited the most promising dielectric properties. 

## 2. Materials and Methods

### 2.1. Synthesis and Reagents

The analytical grade reagents, namely zinc acetate dehydrate (Zn(CH_3_COO)_2_·2H_2_O, 98%), Chromium (III) nitrate nonahydrate (Cr(NO_3_)_3_·9H_2_O, 99%) and tartaric acid, were purchased from Sigma-Aldrich (St. Louis, Missouri, United States) and used as received for ZnCr_2_O_4_ nanoparticle synthesis by the sol-gel auto-combustion method. Tartaric acid was used as a chelating-fuel agent. As in a typical synthesis, the reagents were mixed in their respective molar ratio followed by their dissolution in distilled water. The molar ratio for metal cations Zn^2+^:Cr^3+^ was 1:2, while for chromite: tartaric acid it was 1:3. The resulting dark violet solution obtained after mixing the precursors was stirred at 75 °C until the excess water evaporated. Subsequently, the resulting gel mixture was kept at ambient temperature for up to 24 h to allow for the complete gel formation. Afterwards, the gel mixture was thermally treated in a sand bath while the temperature was increased from 100 °C to 350 °C with a step of 50 °C per hour. The thermal treatment led to an auto-combustion reaction, which was followed by post-synthesis annealing at 500 °C, 700 °C, 800 °C and 900 °C for 5, 7, 9 and 11 h, respectively. The optimization of reaction conditions was achieved by analyzing the samples at each stage of the synthesis process. We systematically investigated the influence of annealing temperature and annealing time on the nanoparticle size and morphology. The samples were kept at the same temperature for 1, 3, and 5 h (see Appendix A from ESI). In addition, on the ZnCr_2_O_4_ nanoparticles annealed at 500 °C, we compared the size evolution for annealing times of 5 h and 21 h. As can be seen in Appendix A from ESI, by increasing the annealing temperature, only a slight increase in the particle’s size was observed in the samples annealed at 500 °C. However, the distinctive octahedral morphology was exclusively observed only in the samples that were maintained at 900 °C for a duration of 11 h.

### 2.2. Preparation of ZnCr_2_O_4_ Nanoparticle Suspensions 

The resulting ZnCr_2_O_4_ nanopowders were processed into nanoparticle-based thin film structures by suspending the nanopowders into toluene (99.5% purity), in which a small amount (0.45 mg/mL) of elastomer butadiene styrene (SBS) from Merck was dissolved. The suspension was sonicated to ensure the proper dispersion of the nanoparticles. After sonication, the suspension was left undisturbed for 15 min, allowing excess particles to precipitate. Finally, the supernatant was collected and used for the thin film deposition by the drop-casting method. A toluene solution of SBS was used for several reasons. Firstly, the evaporation of toluene is relatively slow to minimize the number of cracks in the thin film. In addition, SBS exerts a complementary effect in minimizing the number of cracks, as it is an elastomer, which will minimize the stress during the formation of the films, thereby improving their quality. The deposition of ZnCr_2_O_4_ nanoparticles from suspension was directed on silver-coated glass substrate. The prepared thin film was allowed to dry under ambient conditions and used for further device fabrication. 

### 2.3. Characterization of ZnCr_2_O_4_ Nanoparticles, Pellets, Films and Fabricated Devices

The morphological and elemental analyses of both nanoparticles and nanoparticle-based films were performed using a Hitachi SU-70 Field Emission Scanning Electron Microscope (FE-SEM) equipped with an Oxford Instrument EDX-detector. The phase formation and purity of the crystal structure ZnCr_2_O_4_ nanopowders were examined by Raman spectroscopy and powder X-ray diffraction (PXRD). Raman spectra were collected using a Horiba LabRAM HR spectrometer covering the range from 150 to 1100 cm^−1^ for all the samples.

Room temperature powder X-ray diffraction (RT-PXRD) patterns were collected on a PANalytical X’Pert Pro X-ray diffractometer using a Cu anode (λ = 1.54 Å for Kα1 radiation). Powder samples were placed on a zero-background Si holder, and diffraction patterns were collected in the angular range from 10 to 80 degrees in 2θ° with a 0.02°/min step size. Powder X-ray diffraction patterns were processed by using the PANalytical X’PertHighScore Plus software (version 3.0.5) and structure refinement was performed by the Rietveld method using the FullProf program. A six-coefficient polynomial function was used to model the background, whereas the peak shape was described by a pseudo-Voigt function. 

The dielectric properties of the samples and capacitors were measured using a CONCEPT 40 Broadband Dielectric Spectrometer (Novocontrol Technologies GmbH & Co. KG, Montabaur, Germany) equipped with an Alpha-A high performance frequency analyzer. The measurements were conducted in the frequency range from 1 Hz to 10 MHz, respectively, in a closed temperature cell in a nitrogen atmosphere in order to avoid moisture effects. Thin film capacitors were fabricated on glass substrates, and the top and bottom electrodes were patterned using sputtered metal coatings through the Quorum Sputter Coater, Model: Q150TES. Top electrodes were deposited using a mask with a 1.5 mm in diameter. Both Ag electrodes were deposited by sputtering, with a thickness of 20 nm. The thickness of both electrodes and dielectric film were measured by scanning electron microscopy, in cross section, after cryogenic fracturing. The geometrical parameters were used to evaluate the dielectric permittivity through BDS analyses. 

The dielectric properties of ZnCr_2_O_4_ nanoparticle-based pellets were also investigated. In order to increase the pellets’ density and to improve their mechanical properties, polyvinyl alcohol (PVA) was used as binder. Thus, ZnCr_2_O_4_ nanoparticles were mixed with a few drops of 10 wt% polyvinyl alcohol (PVA) aqua solution and subsequently pressed into pellets with a thickness of about 0.4 mm and 5 mm in diameter. Silver electrodes with a thickness of 20 nm were deposited on both sides of the pellets, ensuring good electrical contact.

## 3. Results and Discussion

### 3.1. Raman Spectroscopy

The Raman spectra exhibit well defined peaks at 180, 450, 510, 605 and 685 cm^−1^ corresponding to the ^1^F_2g_, E_g_, ^2^F_2g_, ^3^F_2g_ and A_1g_, respectively active modes of the ZnCr_2_O_4_ spinel (Figure 1). These results are in good agreement with the previously published data, revealing the high crystallinity of the samples and the absence of secondary phases [26,27,28,29]. 

However, it is evident that the nanoparticles treated at 500 °C show only an initial formation of spinel structure because the main peak (A_1g_) located at 684 cm^−1^ is broad and small. The ^1^F_2g_ mode at 182 cm^−1^ is also barely visible, thereby indicating the formation of the ZnCr_2_O_4_ spinel structure. In contrast, the bands located around 348 and 550 cm^−1^ are most likely attributed to the E_g_ and A_1g_ modes of Cr_2_O_3_ [30,31]. This suggests that annealing at 500 °C for 5 h is insufficient for the formation of crystalline ZnCr_2_O_4_ nanoparticles, the chemical reaction being incomplete. Consequently, the samples underwent a second annealing treatment at different temperatures, namely 700, 800 and 900 °C for 7, 9 and 11 h, respectively, to allow the formation of the single phase, crystalline zinc chromite nanopowders. The Raman spectra of the nanopowders also reveal that the peak around 450 cm^−1^, corresponding to the E_g_ mode, is slightly visible and its absorption value increases with temperature. This is to be expected because as the annealing temperature increases, the size of nanoparticles becomes comparable to those in the bulk ZnCr_2_O_4_ crystal. The intensity of the peak corresponding to the E_g_ mode is small in the bulk crystal [26], making it even difficult to identify in nanocrystalline ZnCr_2_O_4_.

### 3.2. Structure Analysis 

The phase purity and crystallinity of the nanopowdered ZnCr_2_O_4_ samples were analyzed by powder X-ray diffraction. The corresponding diffractograms confirm the Raman analysis, showing a similar trend in the formation of spinel type nanoparticles. Figure 2a represents the reference XRD spectrum (JCPDS No. 22-1107) [2], whereas Figure 2b–e correspond to ZnCr_2_O_4_ samples annealed at 500, 700, 800 and 900 °C in air, respectively. 

The sample treated at 500 °C exhibits the presence of a secondary phase corresponding to the Cr_2_O_3_ structure (marked with ★, with the reference pattern of Cr_2_O_3_ [32]). In contrast, all other samples annealed at 700, 800 and 900 °C confirm the formation of the pure ZnCr_2_O_4_ spinel phase. To further investigate the crystal structure and atomic site distribution of the ions in the zinc chromite nanopowders, the powder X-ray diffraction data were fitted using the Rietveld analysis in the cubic system (space group *F*$\bar{d3}$*m*). Figure 3 shows the Rietveld refinement of the ZnCr_2_O_4_ nanopowders annealed in air at 700 °C for 7 h. 

It can be easily observed that the fitted curve matches very well with the experimental one and the positions of the Bragg reflections are very similar to those corresponding to the indexed peaks in the spinel structure. The calculated atomic coordinates, lattice parameter and reliability factors are listed in Table 1. The refined cubic lattice parameter a = 8.327(7) Å matches well with the value of the lattice parameter of the bulk standard material (a = 8.327(5) Å; JCPDS 22-1107). Similar values for the crystallographic parameters have been obtained for the samples annealed at 800 °C and 900 °C, as can be seen in the ESI (See Appendix A from ESI). 

### 3.3. Morphology and Elemental Analysis

The examination of the field-emission scanning electron microscopy (FE-SEM) micrograph presented in Figure 4 suggests a strong dependence of the morphology of as-synthesized ZnCr_2_O_4_ nanoparticles on the annealing temperature, whereby the increase of the treatment temperature promotes the growth of the nanoparticles along with the development of well-defined faces, eventually resulting in the formation of octahedral-like nanoparticles.

Specifically, the detailed analysis of the SEM images revealed that whereas the nanopowders obtained after a post-synthesis annealing at 500 °C for 5 h in air contain shapeless nanoparticles with an average size of 10 ± 2 nm (Figure 4a), subsequent heat treatments of the same sample at 700 °C for 7 h, 800 °C for 9 h and 900 °C for 11 h lead to an increase of the size of the nanoparticles to 30 ± 10 nm, 90 ± 20 nm and 350 ± 32 nm, respectively (Figure 4b–d). EDX analysis was performed on all the samples to determine the elemental distribution and weight percent in the nanoparticles. To this end, the samples were deposited on silver-coated glass, and during analysis the sample that annealed at 500 °C for 5 h in air showed the presence of four chemical elements, namely C, Zn, Cr and O. Other elements such as Si, Ca, Mg and Ag or Pt identified by EDX analysis were not taken into consideration as they originated from the substrate. Further details on the EDX analysis can be found in Appendix A.

In the second step of the thermal treatment at 700 °C in air for 7 h, not only did the particles grow, stabilizing the spinel structure, but also the purity of the sample increased as the amount of carbon residuals in the sample (estimated empirically from the EDX analysis) decreased from 5.4 to 2.2 wt. %.

Appendix A provides information about the EDX elemental map, the EDS spectrum and weight percent elemental distribution in the sample. The sample that annealed at 800 °C for 9 h is free of secondary phases, as no other elements were detected (see Appendix A). Nanoparticles annealed at 900 °C for 11 h were also analyzed by EDX. Similar to the previous annealing step, the main elements contained in the nanoparticles are Zn, Cr and O, respectively (see Appendix A). Moreover, a detailed study was conducted on the morphology and elemental distribution of a single nanoparticle by using the nanopowdered sample annealed at 900 °C in air. A high magnification image of the ZnCr_2_O_4_ nanoparticle revealed the existence of an octahedral shape with a slightly trunked edge (Figure 5a). EDX mapping confirmed that nanoparticle contains three elements, namely Zn, Cr and O (Figure 5b–d), and the quantitative analysis of the spectrum of the detected elements in the ZnCr_2_O_4_ particle matched the expected molar Zn:Cr, ratio (Figure 5e).

The weight percent ratio of Zn:Cr extracted from the EDX analysis are shown in Table 2. The closest Zn:Cr ratio to the expected value (0.628), has been obtained for the samples treated at 900 °C (0.648).

### 3.4. Dielectric Properties of ZnCr_2_O_4_ Nanopowders

The electrical behavior of the ZnCr_2_O_4_ nanopowders was performed by using dielectric spectroscopy. Prior to the analysis, the samples were pressed into pellets by mixing the nanoparticles with polyvinyl alcohol (PVA) aqua solution, used with a binding agent. As expected, the real part of the permittivity increased with the size of nanoparticles, except for the sample annealed at 500 °C in air, which exhibits poor dielectric characteristics, presumably due to the presence of Cr_2_O_3_ as a secondary phase and does not follow the experimentally observed trend of variation of the permittivity. Thus, the lowest value of the dielectric permittivity was observed for single-phase spinel-type nanopowders obtained after heat treatment at 700 °C for 7 h, whereby the dielectric constant increased from ε = 20 at 10 MHz to ε = 600 at 1 Hz for nanoparticles with an average size of about 30 nm (Figure 6a). The sample containing ZnCr_2_O_4_ nanoparticles with an average size of 90 nm that was annealed at 800 °C exhibited dielectric permittivity values ranging from ε = 36 to ε = 300, depending on the measuring frequency. At ν = 10 Hz, it was observed that the dielectric permittivity values for the samples annealed at 700 and 800 °C crossover, which can be explained by the polarization of electrodes as indicated by the higher dielectric loss for the sample annealed at 700 °C, 800 °C and 900 °C (Figure 6b). The low frequency values of the dielectric loss (tanδ) for the samples that annealed at 500, then at 700 and 800 °C, were 3.3; 3.3 and 1.74, respectively, with a dramatic decrease in value for the sample subjected to an additional heat treatment at 900 °C for 11 h. In this latter case the value of the dielectric loss was found to be tanδ = 5 × 10^−4^ at ν = 1 Hz, which increases to tanδ = 2.5 × 10^−3^ for frequencies as high as 10 Mhz.

For intermediate post-synthesis annealing of the as-prepared samples, the loss tangent values were found to decrease with increasing the frequency of the external applied electric field. These results strongly suggest that the shape nanoparticles and the development of well-defined facets leads to a dramatic improvement of their dielectric properties as a result of the successive post-synthesis heat treatments.

All in all, the sample annealed at 900 °C for 11 h appears to be the most promising for device fabrication, as it is a single phase and exhibits a stable dielectric permittivity value across the entire frequency range, with a dielectric constant of ε = 400 at high frequency and ε = 1500 at low frequency, respectively. The highest dielectric permittivity value at low frequency was found for the sample annealed at 500 °C for 5 h in air, which can be attributed to the presence of secondary phases.

### 3.5. Fabrication of ZnCr_2_O_4_ Nanoparticle Based Thin Films

As discussed in the previous section, the sample annealed at 900 °C for 11 h in air was found to be the most promising for the fabrication of nanoparticle-based thin films. Thin film structures were fabricated by dispersing zinc chromite nanoparticles into an SBS solution in toluene with the concentration of 0.1 mg mL^−1^ and the resulting mixture was sonicated for 10 min to ensure the thorough dispersion of the nanoparticles.

Once the ZnCr_2_O_4_ nanoparticles were dispersed in the SBS solution, the solution was left undisturbed for 15 min to allow the precipitation of the largest nanoparticles. The supernatant was then collected after decantation, and thin films were prepared by using the drop-casting method on silver-coated glass substrates. The thickness of the films varied with the volume of the solution applied to the substrate surface, which ranged from 25 to 150 µL, respectively. This procedure allowed the fabrication of nanoparticle-based ZnCr_2_O_4_ films with a thickness of varying from 0.5 to 2 μm (Figure 7).

Both top-view and cross-section SEM images of the film with a thickness of 490 nm obtained from 25 µL volume of the solution revealed that the film was nonuniform and presented voids (Figure 7a). This indicates that a volume of 25 µL of solution is insufficient for forming a continuous film. As seen in Figure 7b–f, upon increasing the volume of the solution to 50, 75, 100, 125 and 150 µL, respectively, the quality of the films improved considerably, although some slight porosity was still observed in all the films fabricated by this method. This suggested that a higher concentration of SBS in the solution was needed to evenly fill the pores with the polymer. Therefore, a suspension of zinc chromite nanoparticles with an SBS concentration of about 0.45 mg mL^−1^ was prepared to obtain nanoparticle-based ZnCr_2_O_4_ thin films for device fabrication.

### 3.6. Device Fabrication

Nanoparticulate ZnCr_2_O_4_ thin film structures were deposited on silver-coated glass substrates and subsequently incorporated as the dielectric layer into capacitors. The ZnCr_2_O_4_-SBS solution was prepared using spinel nanopowders annealed at 900 °C in air for 11 h. The solution was drop cast on the substrate followed by the evaporation of the solvent and the formation of the thin film structures. The top view SEM image of ZnCr_2_O_4_-SBS film suggests that the nanoparticles have been uniformly mixed with the SBS polymer (Figure 8a). Cross-section analysis (Figure 8b) revealed that the film has a thickness of approximately 2 µm, which was subsequently used to calculate the permittivity of the capacitor.

Furthermore, the EDX mapping analysis confirmed the presence of elemental Zn, Cr and O in a molar ratio corresponding to ZnCr_2_O_4_ and C from SBS, elements which were found to be uniformly distributed throughout the film. As can been seen in Figure 8c, the bottom electrode was also clearly visible. After the formation of the nanoparticle-based films, top electrodes were sputtered on these films in a planar capacitor geometry (Figure 8d). EDX analysis was therefore conducted to investigate the chemical composition of the capacitor. Layered EDX maps of the elemental composition of the planar capacitors are presented in Figure 8e illustrating the elemental distribution across different layers of the device. Capacitance and dielectric permittivity measurements of the capacitors were performed within the frequency range of ν = 1 Hz to ν = 10 MHz. The values of the capacitance of the device were found to vary from 10^−11^ F at high frequencies to 10^−9^ F at low frequencies (Figure 8f), whereas its permittivity decreased from about ε = 1000 at low frequencies to ε = 5 ν = 10^4^ Hz, respectively, reaching a plateau in the high frequency range (Figure 8g).

At high frequencies, the dielectric constant values of the thin films are influenced by polarization processes [33,34]. Specifically, at lower frequencies the dipoles tend to align themselves with the oscillating electric field, whereas when the frequency increases, it becomes more challenging for the dipoles to align rapidly with the polarity of the electric field. At lower frequencies, the dielectric response of the sample is dominated by space charge polarization due to the accumulation of charges at the boundaries between the spinel zinc chromite nanoparticles and between the nanoparticles and the SBS elastomer, thereby leading to higher values of the dielectric constant. The onset of observed Debye dielectric dispersion of the dielectric constant for the ZnCr_2_O_4_-SBS-based capacitor could be tentatively explained by Koop’s theory, which is based on the Maxwell–Wagner model [35,36]. The dielectric performance of the nanoparticle-based ZnCr_2_O_4_-SBS planar capacitors is comparable and even superior to that of spinel ferrite-based devices reported in the literature. For example, CoFe_2_O_4_ thin film materials were reported to have a value of dielectric permittivity that varies from ε = 117 to about ε = 100 at frequencies between 20 Hz and 1 MHz [37]. Bellino and colleagues reported values of the dielectric constant for MgFe_2_O_4_ thin films, which decreased from ε = 140 to ε = 1 when the measuring frequency increased from ν = 20 Hz to ν = 2 MHz, respectively [33], whereas others reported values below ε = 14 for MgAl_2_O_4_ [38], NiFe_2-x_Cr_x_O_4_ [39], Mg_0.6_Cu_0.2_Ni_0.2_Cr_2_O_4_ [39], and CoFe_2_O_4_ thin films [40].

## 4. Conclusions

To conclude, ZnCr_2_O_4_ nanopowders with a high dielectric constant and well-defined morphology of the constituting nanoparticles have been successfully synthesized by a simple, yet highly reliable solution-based method. The post-synthesis annealing temperature was found to play a crucial role on the morphology of the nanoparticles and, implicitly, the dielectric characteristics of the ZnCr_2_O_4_ nanopowders. Specifically, the annealing temperatures, which resulted in the formation of single-phase ZnCr_2_O_4_ nanopowders, range between 700 °C and 900 °C and the annealing time of 7 h, whereas lower annealing temperatures resulted in the formation of a secondary phase, identified as Cr_2_O_3_. However, a well-defined octahedral morphology of the nanoparticles is obtained at an annealing temperature of 900 °C for at least 11 h. Overall, this work provides valuable insights into the synthesis, characterization, and possibility for the use of nanostructured spinel chromites in advanced capacitor technologies.

## Figures and Tables

**Figure 1 micromachines-14-01759-f001:**
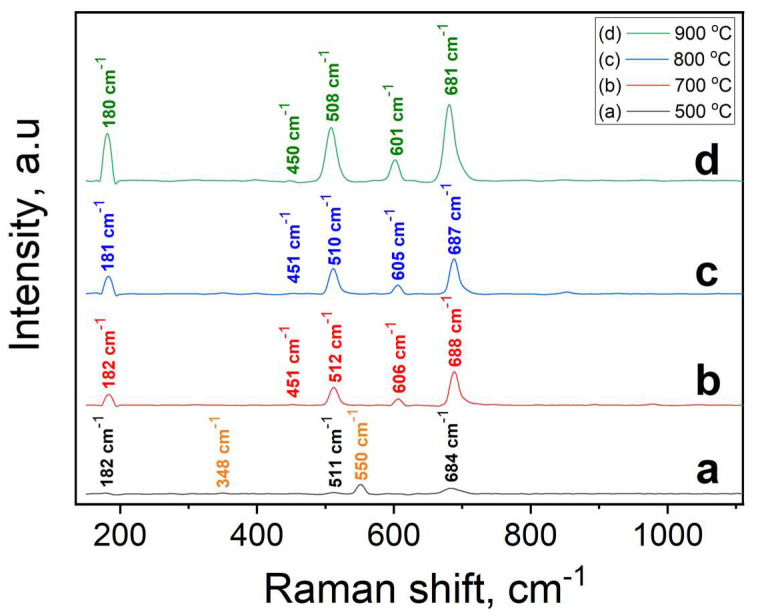
Raman spectra of ZnCr_2_O_4_ samples after annealing under 500 °C (**a**), 700 °C (**b**), 800 °C (**c**) and 900 °C (**d**) for 5, 7, 9 and 11 h, respectively.

**Figure 2 micromachines-14-01759-f002:**
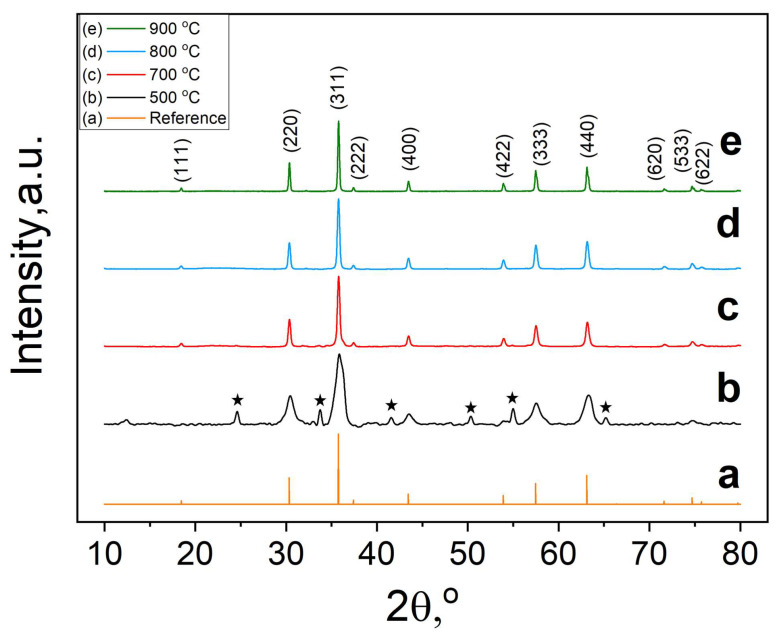
XRD diffractograms of ZnCr_2_O_4_ samples: reference pattern (**a**), and powder XRD patterns of nanoparticles annealed at 500, 700, 800 and 900 °C (**b**–**e**), respectively. The reference pattern of Cr_2_O_3_ are marked with ★.

**Figure 3 micromachines-14-01759-f003:**
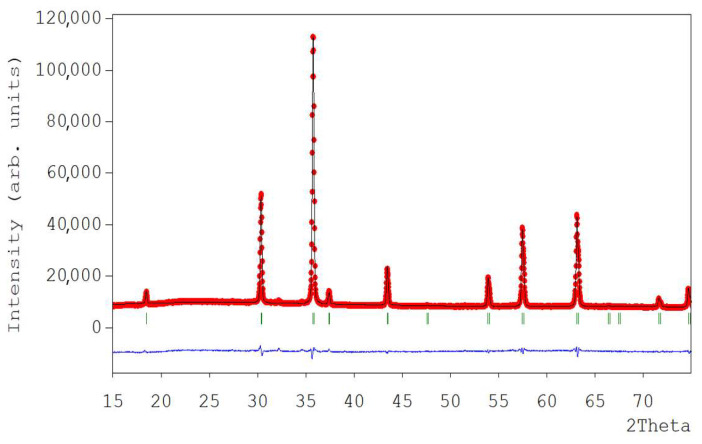
Structure refinement by Rietveld method of nanostructured ZnCr_2_O_4_ samples annealed in air at 700 °C for 7 h (**Red curve**). The blue curve represents the difference I_obs_–I_calc_ pattern, whereas the green vertical bars represent the positions of the Bragg reflections.

**Figure 4 micromachines-14-01759-f004:**
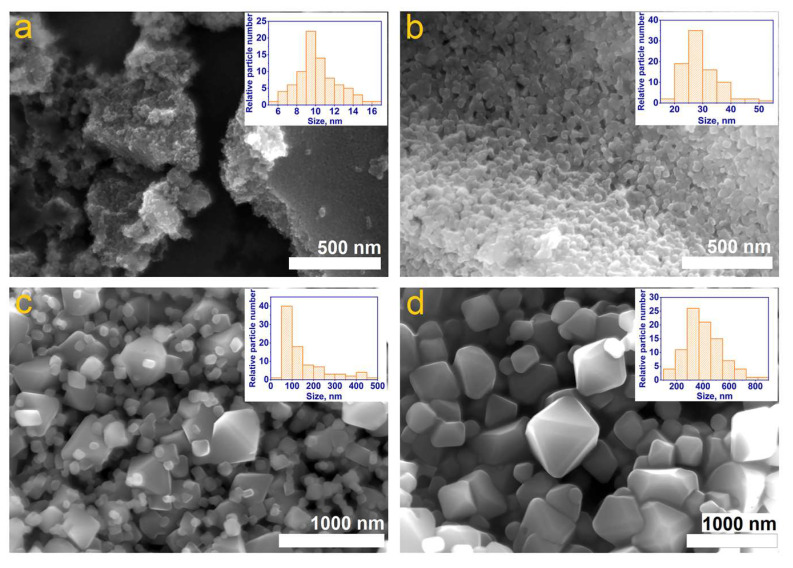
FE–SEM micrographs recorded on ZnCr_2_O_4_ after thermal treatment at 500 °C (**a**), 700 °C (**b**), 800 °C (**c**) and 900 °C (**d**), respectively.

**Figure 5 micromachines-14-01759-f005:**
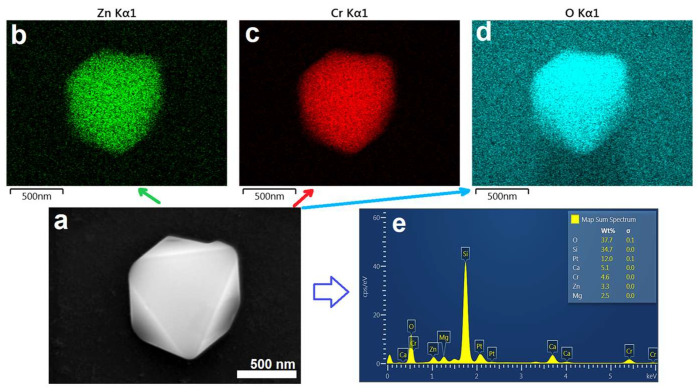
FE–SEM image of an octahedral single particle from a sample annealed at 900 °C (**a**), elemental mapping (**b**–**d**) and EDX spectrum with weight distribution of elements (**e**).

**Figure 6 micromachines-14-01759-f006:**
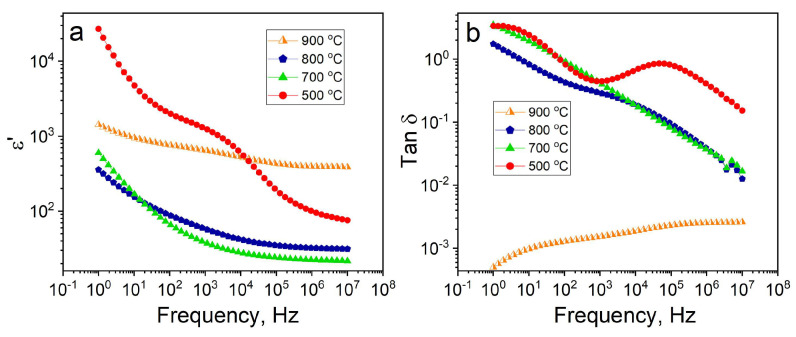
Frequency dependence of the real part of dielectric permittivity (**a**) and Tan δ (**b**), recorded on nanoparticles annealed at various temperatures mixed with PVA binding agent measured at room temperature.

**Figure 7 micromachines-14-01759-f007:**
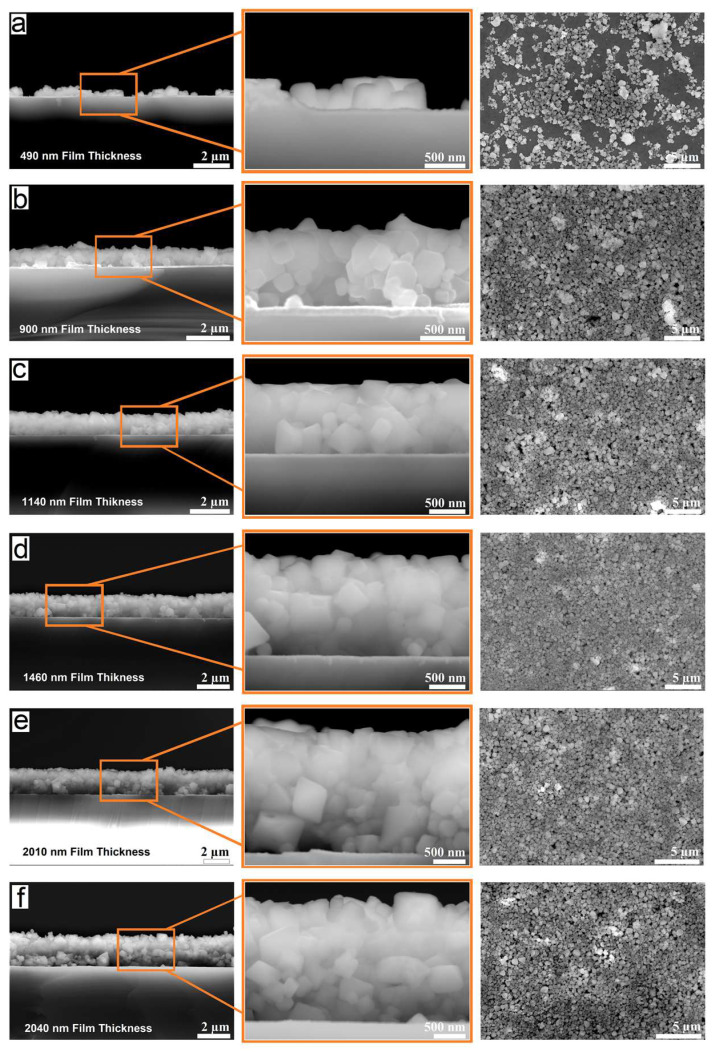
Cross section and top view FE-SEM images of nanoparticulate ZnCr_2_O_4_ thin films with different thicknesses: (**a**) 490 nm; (**b**) 900nm; (**c**) 1140 nm; (**d**) 1460 nm; (**e**) 2010 nm and (**f**) 2040 nm, respectively.

**Figure 8 micromachines-14-01759-f008:**
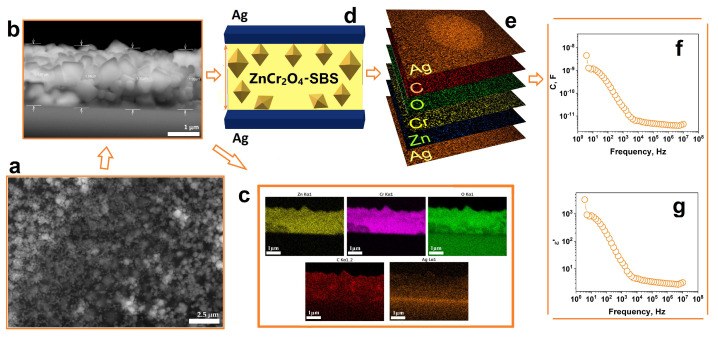
FE–SEM images of thin film of ZnCr_2_O_4_–SBS nanoparticles top view (**a**); cross–section SEM image (**b**), EDX elemental mapping of cross–section (**c**), illustration of produced high–k capacitor (**d**), EDX elemental maps of capacitor (**e**), graph of capacitor capacity (**f**) and dielectric constant (**g**).

**Table 1 micromachines-14-01759-t001:** O_4_ nanopowder annealed at 700 °C in air for 7 h.

Phase	Lattice Parameters (Å)	Atomic Coordinates			Occupancy	B_iso_	R Factors
ZnCr_2_O_4_	a = b = c	Ion	x = y = z	Wyckoff			R_p_ = 9.57%; R_wp_ = 7.15%, R_exp_ = 2.42, χ^2^ = 2.76
	8.327(7)	Zn^2+^	0.125	8a	1.0	0.72	
		Cr^3+^	0.500	16d	1.0	0.45	
		O^2−^	0.259(8)	32e	1.0	0.79	

**Table 2 micromachines-14-01759-t002:** 700, 800 and 900 °C for 5, 7, 9 and 11 h, respectively.

Element	(500 °C) wt%	(700 °C) wt%	(800 °C) wt%	(900 °C) wt%
Zn	3.1	10.3	2.0	3.5
Cr	4.0	14.7	2.8	5.4

## Data Availability

Data are available from the authors on reasonable request.

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
