# Peer review of "Fabrication and Characterization of Dielectric ZnCr_2_O_4_ Nanopowders and Thin Films for Parallel-Plate Capacitor Applications"

_micromachines, 2023, doi:10.3390/mi14091759_

Round 1
Reviewer 1 Report
The authors present an interesting study on the crystal growth and shaping upon thermal treatment and the effect on dielectric properties. The manuscript raised some questions and comments which is found below.
In my opinion, the beginning of the introduction should be a bit more focused on why this topis is of interest rather than stating 20 references that showed different uses of spinel phases followed by 23 references on applications of ZnCr2O4.
Why was the heat-treatment duration prolonged when the temperature was increased? Both temperature and time will affect the crystal-restructuring. Why not keep either temperature or time constant to show the actual influence of one of the parameters? In this regards, it could be interesting to add 3 new samples at 700 or 900 C heat-treated for shorter time (or 500 C at 11 hours) to prove the influence of either one of the parameters changed upon heat treatment. Could the secondary Co3O4 phase at 500 C be eliminated by prolonging the heat-treatment time?
The Rietveld refinement should be described in the Methods section.
In line 212, it is stated that Cr2O3 is marked with an * in the XRD pattern, however, the * is not shown in the figure.
Why is Rietveld refinement only shown for one sample (700 C)? Based on Figure 2, it seems that the peaks sharpen with increasing temperature resulting in a more defined crystal structure. I suggest to add refinement data for all temperatures and discuss the differences.
I suggest to add a table or at least describe in the main manuscript with the ratios of Zn and Cr based on EDX results.
Author Response
We thank the reviewer for his/her comments and suggestions. Please see the attachment.

Reviewer 2 Report
In this paper, shape-controlled synthesis of dielectric ZnCr2O4 is reported. The samples and preparation are clearly addressed. Data with good structural logic is reported. Temperature - morphology relationship is systematically studied. There are some minor questions regarding experimental details:
1. PVA is used for BDS measurements, while SBS with toluene is used for thin film casting. Which solvent was used for dissolving PVA? Why use PVA instead of SBS for BDS testing?
2.Moisture will have effects on the dielecrtric measurements. Please address (if any) drying procedures.
3. Fig. 6, the weird tandelta curve of 900 °C might because of loose contact.
4. Thickness variation is vital for dielectric measurements, no matter for pressed pellets or casted thin films. Please describe the diameter and thickness with variations in experiment preparation part.
5. Similar thermal annealled structure-dielectric properties relationship was published on BTO particles, with discussions and details sample preparations. Might be a good reference to compare.
https://doi.org/10.1016/j.cej.2023.142490
https://doi.org/10.1039/D3NR00350G
Author Response

(The authors gave the same response as above.)

Reviewer 3 Report
The article is devoted to an important and relevant topic and may be of interest to a wide range of readers. There is one important note to the article, which does not allow publishing the article "as is". According to the authors, the purpose of the article is to study the structure, morphology and dielectric properties of ZnCr2O4 depending on the annealing temperature of the material. However, the authors compare the properties of samples that differ from each other in both annealing temperature and annealing time. Thus, in a series of experiments, two important parameters change simultaneously for each sample, and each of the parameters can affect the final properties. In the conclusions, the authors attribute all changes in properties to the annealing temperature, without considering the possible effect of the annealing duration.
The second remark is not essential, but it is important for a general understanding of the results. In Figure 2, there is no asterisk marking mentioned in the text.
Author Response

(The authors gave the same response as above.)

Reviewer 4 Report
The article presents the results of the experiments with the zinc chromite nanopowder. Grain sizes of the powder together with other corresponding structure properties were demonstrated to correlate with annealing processing parameters. Dielectric properties were investigated by parallel-plate capacitor structure formation and its electric measurements.
The relevance of the topic is justified enough by the modern references. The work has a soundness and is good presented, so it can be published after some minor revisions and corrections.
line 169. "Properties" may be?
lines 191-194. How do you explain that the duration of annealing is increased with the annealing temperature? Why the hotter is longer?
line 212. As I can see, nothing is marked with a star in the Figure 2.
line 305. "Increases" may be?
And some comments on the title. I believe that the authors may consider to remove the word "planar" from the title, because it can make readers confused. In my experience the term "planar capacitor" refers to the capacitor with the plates lying in the same plane (see for example [1-3]). However in your work a common parallel-plate capacitor is investigated. I suggest you to consider the title modification.
Reviwer references:
1. V. Laur et al., "Microwave study of tunable planar capacitors using mn-doped ba0.6sr0.4tio3 ceramics," in IEEE Transactions on Ultrasonics, Ferroelectrics, and Frequency Control, vol. 56, no. 11, pp. 2363-2369, November 2009, doi: 10.1109/TUFFC.2009.1324.
2. Outzourhit, A., Trefny, J. U., Kito, T., Yarar, B., Naziripour, A., & Hermann, A. M. (1995). Fabrication and characterization of Ba1− xSr1− xTiO3 tunable thin film capacitors. Thin Solid Films, 259(2), 218-224.
3. Xiaohui Hu Wuqiang Yang, (2010),"Planar capacitive sensors – designs and applications", Sensor Review, Vol. 30 Iss 1 pp. 24 -
39
Author Response

(The authors gave the same response as above.)

Round 2
Reviewer 1 Report
The authors generally improved the manuscript.
I would still suggest to add Rietveld refinement data for different synthesis conditions to clearly show the difference in the crystal structure with increasing temperature and duration. E.g. add data for 800 C and 900 C to Table 1.
Reviewer 3 Report
On the reviewer`s opinion, the main problem of the manuscript has not been solved in the updated version. Figures 1, 2, and 6 compare samples that differ from each other in both temperature and annealing time. Both of these technological parameters certainly affect both the structure and the electrical characteristics of the samples. The authors conclude that the most promising characteristics are shown by samples annealed at 900 degrees for 11 hours. At the same time, other annealing times at this temperature are not considered. To consider these parameters (900 oC /11 h) optimal, it is necessary to compare, for example, diffractograms of samples annealed at a given temperature for different times, or, better, to compare their dielectric permittivity.
Round 3
Reviewer 1 Report
The authors provided the requested data to the manuscript, and can now be published.
There are some misspellings that should be fixed.
Reviewer 3 Report
The authors took into account all the comments of the reviewer and improved the manuscript. The article can be published in this form.